# Retrieval & Fine-Tuning for In-Context Tabular Models

**Valentin Thomas** [* 1]  **Junwei Ma** [* 1]  **Rasa Hosseinzadeh** [1]
**Keyvan Golestan** [1]  **Guangwei Yu** [1]  **Maksims Volkovs** [1]  **Anthony Caterini** [1]

## Abstract

Tabular data is a pervasive modality spanning a wide range of domains, and the inherent diversity poses a considerable challenge for deep learning. Recent advancements using transformer-based in-context learning have shown promise on smaller and less complex datasets, but have struggled to scale to larger and more complex ones. To address this limitation, we propose a combination of retrieval and fine-tuning: we can adapt the transformer to a local subset of the data by collecting nearest neighbours, and then perform task-specific fine-tuning with this retrieved set of neighbours in context. Using TabPFN as the base model – currently the best tabular in-context learner – and applying our retrieval and fine-tuning scheme on top results in what we call a locally-calibrated PFN, or LoCalPFN. We conduct extensive evaluation on 95 datasets curated by TabZilla from OpenML, upon which we establish a new state-of-the-art with LoCalPFN – even with respect to tuned tree-based models. Notably, we show a significant boost in performance compared to the base in-context model, demonstrating the efficacy of our approach and advancing the frontier of deep learning in tabular data.

## 1. Introduction

Tabular data is the most pervasive modality for practical problems in data science, spanning across a wide variety of domains including finance, healthcare, and science (Benjelloun et al., 2020; Ulmer et al., 2020; Clements et al., 2020; Tang et al., 2020; Urban & Gates, 2021). The diversity and heterogeneity of tabular data pose great challenges for deep learning approaches (Grinsztajn et al., 2022), unlike modalities such as text and image

in which neural networks can be designed to specifically exploit inductive biases underlying the data (Borisov et al., 2022). As such, obtaining a performant neural network on a particular tabular data task often results in expensive iterations of training and hyperparameter tuning. Meanwhile, tree-based methods such as XGBoost (Chen & Guestrin, 2016) and CatBoost (Prokhorenkova et al., 2018) have proven to be more robust to the inherent challenges of tabular data, and thus have remained the dominant approach for this setting (Grinsztajn et al., 2022; Shwartz-Ziv & Armon, 2022; Borisov et al., 2022). Yet recently, there has been progress made with transformers and In-Context Learning (ICL): one such example is TabPFN (Hollmann et al., 2023), which is trained using a prior-fitting procedure (Müller et al., 2022) that exposes the network to millions of possible data-generating processes, thus taking a step towards encapsulating the heterogeneity of tabular data. Such approaches differ from classical algorithms in that they process entirely new datasets in a single forward pass and obviate the need for training and hyperparameter tuning.

Despite the promise of transformer-based ICL methods in the tabular setting – particularly on smaller datasets – scaling remains an issue: memory scales *quadratically* in the size of the context. This limits performance when the entire dataset cannot fit into memory, and contrasts with classical algorithms that tend to improve as the amount of available data increases. In addition to this, and as depicted in Figure 1, TabPFN in particular can struggle with underfitting as dataset *complexity* increases, even when the entire dataset fits into the context; we observe this shortcoming in real datasets as well, and suspect this could apply to any ICL-based model for tabular data.

To improve the scaling of tabular ICL methods in both dataset size and complexity, we draw on two techniques that have been incredibly successful in foundational large language models: retrieval (Lewis et al., 2020) and fine-tuning (Bommasani et al., 2021). On the retrieval side, we use the $k$-Nearest Neighbours ($k$NN) of a given query point as the context for classification; modifying the context in this way empirically allows for both enhanced processing of larger datasets and more complex decision boundaries. We also fine-tune end-to-end for each task, using an approximate neighbour scheme to

---

*Equal contribution [1]Layer 6 AI, Toronto, Canada. Correspondence to: Valentin Thomas <valentin.t@layer6.ai>, Junwei Ma <jeremy@layer6.ai>.

*Proceedings of the 1st Workshop on In-Context Learning at the 41st International Conference on Machine Learning*, Vienna, Austria. 2024. Copyright 2024 by the author(s).

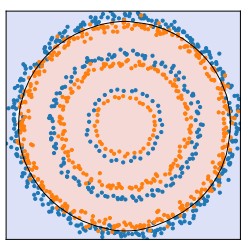
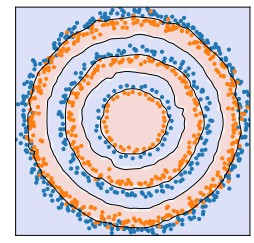
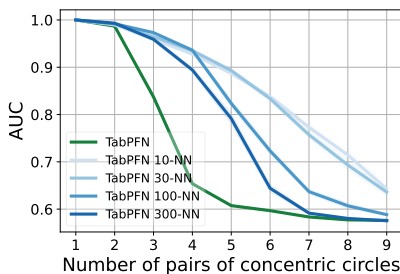

(a) Vanilla TabPFN, full context      (b) TabPFN-$k$NN, $k = 100$      (c) Performance vs. Complexity

Figure 1: a) TabPFN – even when using the entire training data as context – underfits and cannot classify patterns such as three pairs of concentric circles of two classes. Decision boundaries are in black and shaded areas show the predicted class. b) Applying an adaptive local context for each point using its $k$ nearest neighbours can easily solve this problem. c) We observe that this approach is robust to the numbers of neighbours used ($k$) even when the dataset complexity increases and always performs better than vanilla TabPFN using full context ($N = 1000$). Each point is averaged over 25 seeds.

facilitate backpropagation, and demonstrate significant performance gains beyond just $k$NN. We named our model Locally-Calibrated PFN – or LoCalPFN for short – to represent the addition of retrieval and fine-tuning on top of a base TabPFN model, although this idea should naturally transfer to potential future ICL-based tabular foundation models as well (van Breugel & van der Schaar, 2024). We demonstrate that LoCalPFN is state-of-the-art against both neural approaches and well-tuned tree-based techniques across a 95-dataset benchmark from TabZilla (McElfresh et al., 2023). We summarize our contributions below:

1. Provide insights into TabPFN – the current state-of-the-art tabular ICL transformer-based framework – and analyze how its performance scales across several axes in both synthetic and real datasets. We identify a failure to scale in both dataset size and complexity.

2. Propose LoCalPFN to address the scaling failures mentioned above, using a combination of retrieval and fine-tuning to allow for more effective use of the context.

3. Show LoCalPFN compares favourably to strong baselines on a large variety of datasets through extensive experimentation, analysis, and ablation.

## 2. Improving Tabular In-Context Learning with Retrieval and Fine-Tuning

In this section, we describe ICL applied to tabular data – in particular TabPFN – and the limitations of such an approach. Then, we present our contributions where we treat the in-context learner as a base model on top of which retrieval and fine-tuning are applied.

### 2.1. Preliminaries

Our method generally applies to in-context learners, specifically for classification tasks on tabular data. While, at the time of writing the only successful model of that type is TabPFN (Hollmann et al., 2023), we expect other such base models to be published in the future. TabPFN is trained using a prior-fitting procedure (Müller et al., 2022) where a large number of synthetic datasets are generated using randomly initialized neural networks. This approach trains an underlying transformer-based network on various generative processes designed to simulate the diverse interrelations that exist among the features of realistic tabular datasets.

After the prior-fitting procedure, the learned TabPFN model ingests an entire training dataset $\mathcal{D}_{\text{train}} \triangleq \{(x^i_{\text{train}}, y^i_{\text{train}})\}^N_{i=1}$ consisting of feature-label pairs $x^i_{\text{train}} \in \mathbb{R}^D$ and $y^i_{\text{train}} \in \{1, \ldots, C\}$ for $i \in \{1, \ldots, N\}$, along with features of a query point $x_{\text{qy}}$ (potentially in a batch), and outputs a distribution over labels $y_{\text{qy}} \in \{1, \ldots, C\}$. Specifically, denoting the TabPFN network (outputting logits) as $f$, the resulting posterior predictive distribution is modelled by:

$$p_\theta(y_{\text{qy}} \mid x_{\text{qy}}, \mathcal{D}_{\text{train}}) = \frac{\exp(f_\theta(x_{\text{qy}}, \mathcal{D}_{\text{train}})[y_{\text{qy}}])}{\sum_{c=1}^C \exp(f_\theta(x_{\text{qy}}, \mathcal{D}_{\text{train}})[c])}, \quad (1)$$

where $[\cdot]$ denotes the vector indexing operation.

Contrary to classical machine learning methods which are trained on one dataset and then evaluated on the same distribution, TabPFN has been shown to be able to perform classification on a wide range of tasks without training, thanks to its diverse prior-fitting procedure. This makes it one of the rare foundation models for tabular data. Key to this is the ICL ability of TabPFN: by using various training *examples* as context, analogous to how transformers on language use the preceding *tokens* as context, TabPFN can classify new query points in a single forward pass.

## 2.2. What is a Good Context for Tabular Data?

The quadratic growth of memory usage with context length in transformers presents a challenge: the number of support examples we can use is limited. For instance, while TabPFN performs best on small and simple datasets, where the entire training set fits within the context, it is unclear how to best use TabPFN for large and complex datasets. Naïvely, we might consider a random subsample of the training data as context (McElfresh et al., 2023; Feuer et al., 2024). However, Ma et al. (2024) show that this method does not scale either and observe a drop in performance as the dataset size increases.

Given these limitations, it is natural to ask *"What constitutes a good context for tabular data?"*. This topic has been thoroughly researched in natural language processing, which resulted in various techniques for prompt engineering. The situation is more complicated in the tabular domain, as there is no natural order to tabular data as opposed to the natural order of the words in language.

Specifically for TabPFN, some attempts have been made to use a summary of the dataset as context, through either $k$-means centroids (Feuer et al., 2024) or direct prompt optimization (Feuer et al., 2024; Ma et al., 2024). Yet in either case the flexibility of the method is limited by the use of a single context for all query points. Instead, we propose a different approach here, where we use a local context tailored to each individual point we wish to classify. For tabular data, we hypothesize that the most critical information to classify a query point $x_{qy}$ is contained in its vicinity. Extensive evaluations (McElfresh et al., 2023) support this fact by showing that a simple $k$NN classifier can rival modern deep architectures designed for tabular data, such as TabNet (Arik & Pfister, 2021) and VIME (Yoon et al., 2020). We thus believe that using nearby points as context is a good inductive bias for tabular data classification.

## 2.3. Better Expressivity and Scaling with local information

To do this, the first step is to replace the *global* context by a *local* context, i.e., with $k$NN$(x_{qy})$ as the $k$-nearest neighbours of the query $x_{qy}$ in the training data $\mathcal{D}_{train}$, we replace equation 1 by

$$p_\theta(y \mid x_{qy}, \mathcal{D}_{train}) = \frac{\exp(f_\theta(x_{qy}, k\text{NN}(x_{qy}))[y])}{\sum_{c=1}^{C} \exp(f_\theta(x_{qy}, k\text{NN}(x_{qy}))[c])}. \quad (2)$$

**Better Expressivity** It is well known that in $k$NN regression and classification, the number of neighbours $k$ controls the bias/variance trade-off and as such the expressivity of the model. More precisely, large $k$ tends to "oversmooth" and suffer from high bias/underfitting, while small $k$ enables more complex decision boundaries but can suffer from more

variance/overfitting (Hastie et al., 2009). We show that this phenomenon is still true for transformers, beyond the simple $k$NN classifier, in Figure 1. We generate datasets of size $N = 1000$ so that it can be used as context by TabPFN without subsampling. As we increase the complexity of the dataset, measured by the number of concentric circles in this case, TabPFN fails to accurately classify (e.g., for 3 pairs of circles in (a) and more generally in (c)). Retrieving fewer samples ($k = 10, 30, 100,$ or $300$) for each query point using its $k$-nearest neighbours from the training data leads to large improvements in AUC over TabPFN as the complexity of the data increases ((b) and (c)). Note that $k = 1000$ would correspond to using all samples as context, and thus is equivalent to vanilla TabPFN. As such there is a continuum between TabPFN using the full dataset as context and our local context method using $k$NN, which we call TabPFN-$k$NN.

While Figure 1 is on toy synthetic data, we believe this result remains surprising: *a priori*, we would expect a 25-million-parameter model (TabPFN) to be able to learn a few circles, even with just ICL. Meanwhile, we believe that using local contexts allows TabPFN to fit more complex patterns, such as the three circles of Figure 1, in the same way that using local linear regression enables more expressive (and in that case nonlinear) decision boundaries (Cleveland & Devlin, 1988; Hastie, 2017).

**Better Scaling** Using a local context has another benefit: it allows our method's performance to scale with the training dataset size. In machine learning, it is generally expected that the performance of an algorithm improves as the training set size $N$ increases, since the empirical risk converges to the expected risk (Vapnik, 2013). However, ICL-based methods (such as TabPFN) that require subsampling when the maximum context length is smaller than $N$ do not scale with $N$. TabPFN-$k$NN, on the other hand, can still benefit from larger training set sizes $N$ even when the number of neighbours $k$ is much smaller than $N$, as the search is performed over the whole training set. We demonstrate this fact in Figure 2 for three real datasets. While the exact patterns in the loss curves differ, we observe a similar trend across many datasets, where the benefits of using TabPFN-$k$NN grow as the dataset becomes larger. In Figure 10 we provide more detailed figures which include training loss.

## 2.4. Efficient End-to-End Fine-Tuning With Retrieval

In addition to retrieval, we fine-tune the model end-to-end on each dataset to further improve performance, as is common in Retrieval-Augmented Generation (RAG) (Lewis et al., 2020). However, naïve fine-tuning is not computationally efficient. Transformer-based in-context models work with inputs of shape $(B, L_{ctx} + L_{qy}, d)$ where $B$ is the batch size, $L_{ctx}$ and $L_{qy}$ are the context and query lengths, and $d$ is the embedding dimension. TabPFN uses only one fixed con-

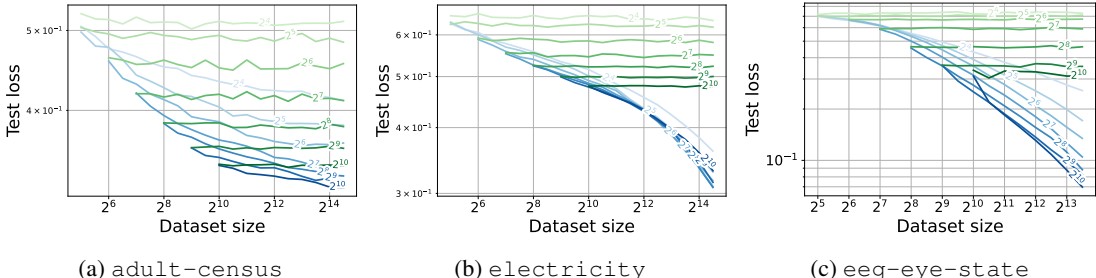

(a) `adult-census`  (b) `electricity`  (c) `eeg-eye-state`

Figure 2: Example of the behaviour of TabPFN and TabPFN-$k$NN as we vary the dataset size and the context length for three large datasets. TabPFN is in shades of green and TabPFN-$k$NN is in shades of blue. The opacity represents the context length used (also labelled on each line). It corresponds to random training samples for TabPFN and nearest neighbours for TabPFN-$k$NN. TabPFN is limited by context size and cannot make efficient use of larger datasets. While for context length = dataset size ($k = N$) TabPFN and TabPFN-$k$NN have the same performance, TabPFN-$k$NN can leverage larger datasets with $k$NN-based contexts and shows improvements, often even for lower context lengths. Each point on this plot is the average of 100 random resamplings of the data.

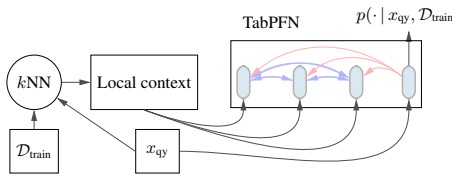

Figure 3: Overall architecture of LoCalPFN. During inference, for each query $x_{\mathrm{qy}}$, we compute its $k$NNs and use them as context.

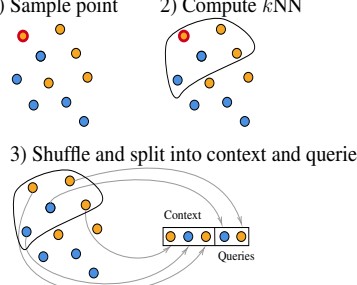

Figure 4: Efficient local context computation for fine-tuning.

text for all points, with $B = 1$, $L_{\mathrm{ctx}}$ the training dataset size (or maximum context length if too large), and $L_{\mathrm{qy}} = N_{\mathrm{qy}}$ the number of points to classify. Contrary to text, there is no auto-regressive attention mask: the context examples all attend to each other (blue arrows on Figure 3) while the queries only attend to the context and not to each other (red arrows on Figure 3). Therefore, the predicted classes can be computed in parallel and at a reduced memory footprint.

By comparison, when using a local context with exact neighbours, the context is no longer shared, and therefore the batch dimension must be used for queries: the input has shape $B = N_{\mathrm{qy}}$, $L_{\mathrm{ctx}} = k$ – the number of neighbours – and $L_{\mathrm{qy}} = 1$, since the queries use distinct contexts. This is significantly less efficient than the inference performed by TabPFN, which both requires much less memory, and also allows the queries to be processed in parallel. Therefore, our main limitation is in fact the forward and backward passes when using exact neighbors, unlike most applications where retrieval is the bottleneck. As such, most common approximate $k$NN methods cannot address this issue. Instead, to improve computational efficiency during the end-to-end fine-tuning, we opt for a simple neighbour approximation technique wherein many queries share the same context. An

illustration of the method is provided in Section 2.4 for a single batch dimension. More generally, let us assume that we want to pass gradients on $N_{\mathrm{qy}}$ examples at once, using a context length of $L_{\mathrm{ctx}}$. We propose to only use $B$ different contexts, which we will use to classify $N_{\mathrm{qy}}/B$ samples each: First, $B$ training examples are sampled. Then, their individual $k$NN search is performed with $k = L_{\mathrm{ctx}} + L_{\mathrm{qy}}$ for $L_{\mathrm{qy}} = N_{\mathrm{qy}}/B$. Finally, those batches of $k$ samples are shuffled and split into a context vector of length $L_{\mathrm{ctx}}$, and a query vector of length $L_{\mathrm{qy}}$, constructing the input vector of size $(B, L_{\mathrm{ctx}} + L_{\mathrm{qy}}, d)$. This allows us to efficiently trade-off accuracy of the neighbours versus computational complexity: with lower $B$ we share contexts between many points but this comes at the cost of an approximation in the $k$NN search as the notion of neighborhood is not transitive, i.e., the neighbour of your neighbour might not be your neighbour. However each sequence in each batch only contains examples which are in the general vicinity of each other. In practice, we observe that this method does not lead to any significant degradation in performance while allowing much faster training.

# 3. Related work

**Foundational Techniques for Tabular Deep Learners**
Deep learning techniques have historically struggled on tabular data (Grinsztajn et al., 2022), where inductive biases are much harder to capture architecturally (Beyazit et al., 2023) as compared to text or images. The comparative lack of progress on a large foundation model for tabular data (van Breugel & van der Schaar, 2024) is yet more evidence of this. However, recent approaches have successfully begun to leverage foundational ideas to improve performance. For example, Non-Parametric Transformers (Kossen et al., 2021) and SAINT (Somepalli et al., 2021) both combine row-attentive transformer-based backbones with some form of self-supervised pre-training; however, the former is limited by context size (a common theme for naïve ICL-based learners), whereas the latter is not based on ICL and thus does not as easily apply to novel datasets. Models such as RIM (Qin et al., 2021) and TabR (Gorishniy et al., 2024) on the other hand demonstrate how to effectively design tabular deep learners incorporating retrieval modules, but still require costly and brittle rounds of hyperparameter tuning to adapt to any specific dataset. Our approach is meant to target some combination of all these methods: provide ICL-based generalization capabilities, but without limitations on the context size. The retrieval mechanism within TabR itself relies on $k$NN, which is one of the most straightforward and widely used retrieval-based machine learning methods (Hastie et al., 2009). In fact, $k$NN is still being actively studied in the literature, e.g., in Differential Nearest Neighbours Regression (DNNR) (Nader et al., 2022), which aims to make $k$NN differentiable; this showcases the potential of simple methods like $k$NN in different forms, although DNNR tackles a separate scope from our method.

**TabPFN and Extensions**  TabPFN (Hollmann et al., 2023) is a transformer-based in-context learner that has emerged as a popular model for tabular data, demonstrating strong performance on some benchmarks (McElfresh et al., 2023). It uses a prior-fitting process (Müller et al., 2022) allowing for rapid adaptation to new tasks. This strong ability to quickly generalize makes TabPFN somewhat of a foundation model for tabular data (van Breugel & van der Schaar, 2024), from which techniques for generation (Ma et al., 2023) and dataset distillation (Ma et al., 2024) for example can emerge – interpretability is also being studied (Rundel et al., 2024). TuneTables (Feuer et al., 2024) attempts to use tabular sketching (Munteanu & Schwiegelshohn, 2018) to summarize the incoming dataset and more effectively scale TabPFN's context; however, much like Ma et al. (2024), this approach is limited by the use of a single context for all datapoints, as opposed to an adaptive local context. Lastly, den Breejen et al. (2023) is able to show some limited improvements by fine-tuning TabPFN, which we extend here by more closely pairing the retrieval and fine-tuning aspects.

**Links with LLMs**  The idea of pre-training a model on corpora of text prior to fine-tuning has been explored in the Natural Language Processing domain for both classification and generation tasks (Dai & Le, 2015; Howard & Ruder, 2018; Radford et al., 2018). Later iterations refined this idea to train a model and use its in-context learning abilities for new tasks (Brown et al., 2020). This elicited research into prompt engineering to determine what to actually put in a model's context (Nye et al., 2021; Wei et al., 2022). Similar to prompt engineering, to better utilize the model's context, one can search for similar examples from a corpora and use them to facilitate the task; this is known as Retrieval-Augmented Generation (RAG) (Lewis et al., 2020) in the generative context. Other variants of the idea include training jointly with retrieval (Guu et al., 2020; Borgeaud et al., 2022) and augmenting the output of the model with $k$NN via interpolating (Khandelwal et al., 2019). These ideas are analogous to our approach of (i) fine-tuning and retrieving jointly, and (ii) disjoint $k$NN and fine-tuning in our ablations, respectively. LLMs have also been directly applied to tabular data (Dinh et al., 2022; Hegselmann et al., 2023; Fang et al., 2024) however, due to the pre-training of these foundation models on large text corpora, there is the possibility of data leakage, which causes concern with evaluations (Bordt et al., 2024). Note that this is not the case with TabPFN as it has been trained on synthetic data.

# 4. Experiments

## 4.1. Experimental Setup

We evaluate our methods against competitive baselines using 95 out of the 176 datasets from TabZilla (McElfresh et al., 2023), originally sourced from OpenML (Bischl et al., 2021). These datasets originate from diverse sources, including academic research, competitions, government agencies, and corporations. The 95 datasets are filtered from TabZilla to meet TabPFN's architectural requirements by ensuring that each dataset has at most 100 features, at most 10 classes, does not contain NaN values, and has at least one instance per class for each split. The details of the datasets are described in Appendix A.1. We further split the datasets into two subsets: "small" datasets which contain less than 2,000 instances, and "medium/large" which contain at least 2,000 instances (up to 130,064). For each dataset, we use the splits from TabZilla with train-validation-test ratio of 80:10:10. Since TabPFN was trained with a maximum of 1,024 data points as context size, the small datasets are roughly considered in-distribution for TabPFN whereas the large datasets are considered out-of-distribution.

We conduct our experiments using 10-fold cross-validation over all datasets for all methods. For all baselines, we apply 30 rounds of hyperparameter tuning as in McElfresh et al. (2023) and choose the best hyperparameters for each fold

according to its validation AUC. In addition, the TabPFN baseline is reported without further ensembling or transformations, unless otherwise noted. Our methods also build on top of this same TabPFN baseline without further processing. We also compare against TabPFN with transformations in Section 4.4. More details of the baseline models can be found in Appendix A.2.1. We use the `faiss` (Johnson et al., 2019; Douze et al., 2024) library for efficient $k$NN search in our methods; this enables us to harness parallel computation to accelerate the nearest neighbour search. We evaluate our methods TabPFN-$k$NN and LoCalPFN against other models in the following sections. Notably, without further fine-tuning, LoCalPFN is identical to TabPFN-$k$NN. Details of our method are in Appendix A.2.2.

**Note on evaluation and the computation of proper confidence intervals:** While many works evaluate tabular data methods on a small set of datasets and report confidence intervals/standard deviations for those, we choose to evaluate on a large number of datasets in order to have more meaningful results. However, this makes it harder to compute meaningful uncertainty. Agarwal et al. (2021) dealt with a related problem in reinforcement learning; we follow their lead by, for example, reporting the interquartile mean (IQM, i.e., the mean of the middle $50\%$ of scores), and we use their library to compute $95\%$ confidence intervals via stratified bootstrapping.

## 4.2. Main Experiments

As shown in Section 4.2, averaged over 95 datasets, LoCalPFN outperforms all other baselines, with significant improvement over TabPFN itself. Among the 47 small datasets, we found that TabPFN is in fact quite competitive with other methods, similar to what had been reported by McElfresh et al. (2023). Nevertheless, LoCalPFN further improves the performance even in this setting and positions itself as the best method. For the 48 medium/large datasets, TabPFN underperforms the tree-based methods by a wide margin. Simply applying $k$NN on top of TabPFN leads to a drastic performance increase on top of TabPFN. Finally, LoCalPFN further improves on TabPFN-$k$NN, and either performs on par with, or outperforms, all other methods. We also measure the accuracy and F1 score over all datasets and see a similar pattern; those details can be found in Table 6 and Table 7.

**Deep Learning Model Comparisons:** Note that most deep learning baselines are significantly more expensive to train and tune on larger datasets, and as such, most of them could not be run on all datasets (McElfresh et al., 2023). Nevertheless, in Table 5 we compare TabPFN-$k$NN and LoCalPFN to other deep learning based methods on the datasets on which the baselines have been able to run, and show an even larger improvement in performance. The datasets we used for this comparison can be found in Table 4.

## 4.3. Analysis: Scaling with Dataset Size and Complexity

In this section, we further validate that LoCalPFN addresses the scaling problems of TabPFN. We see in Figure 1 and 2 that TabPFN scales badly with both size and complexity; here, we verify this phenomenon in real datasets. While this may appear contradictory to Table 1 of McElfresh et al. (2023), which shows TabPFN excelling on a large benchmark suite, we note that the aforementioned study mostly contained small datasets and thus it did not show the same performance drop-off observed here.

**Scaling with Size** In Figure 5, we report the AUC of different algorithms relative to the AUC of Random Forest for different dataset sizes. We choose relative AUC for clarity as there is no clear correlation between the maximum AUC attainable on a dataset and its size. We see that, compared to the Random Forest baseline, TabPFN's performance drops drastically when the dataset size increases beyond 3,000, indicating poor scaling with $N$. On the other hand, the other methods we report scale more favourably with the dataset size. We also see that LoCalPFN scales favourably compared to the Random Forest baseline, and even outperforms XGBoost for large datasets. Error bars represent the $95\%$ confidence interval.

**Scaling with Complexity** While in Figure 1 we could easily control the complexity of the task, there is no generally accepted measure of complexity for an arbitrary dataset. Here, we propose a simple proxy for complexity: for a given dataset, we measure the difference between the best and worst AUCs of a given set of algorithms, similarly to McElfresh et al. (2023). The rationale is that AUC itself cannot capture complexity, as for instance learning to separate two Gaussians can be done optimally by a linear classifier, but the error rate depends on their variance. In Figure 6, we analyze performance across different levels of this complexity measure. We first calculate the difference in AUC for each dataset using all listed methods in Section 4.2, then we divide the datasets into five quantiles on the $x$-axis, with increasing complexity as we move to the right; on the $y$-axis, we report the mean AUC relative to Random Forest across 10 folds and across the datasets in each bin. We see that TabPFN scales poorly with increasing complexity, and LoCalPFN still outperforms all other methods in the quantiles of higher complexity, demonstrating that its improvements are not just limited to "easy" datasets.

## 4.4. Ablation Studies

In this section, we provide ablation studies on different design choices for LoCalPFN.

**Importance of Joint Retrieval and Fine-Tuning** One could naïvely consider simply fine-tuning the in-context learner on randomly sampled context during training. In

| Algorithm | All | | Small | | Medium/Large | |
|---|---|---|---|---|---|---|
| | IQM AUC | Mean AUC | IQM AUC | Mean AUC | IQM AUC | Mean AUC |
| $k$NN | 0.843 [0.838-0.847] | 0.812 [0.808-0.816] | 0.807 [0.798-0.816] | 0.781 [0.772-0.789] | 0.882 [0.880-0.884] | 0.848 [0.847-0.850] |
| TabPFN | 0.917 [0.914-0.919] | 0.867 [0.864-0.870] | 0.898 [0.892-0.904] | 0.849 [0.843-0.856] | 0.927 [0.925-0.929] | 0.884 [0.883-0.885] |
| TabPFN 3k | 0.924 [0.922-0.927] | 0.873 [0.869-0.876] | 0.903 [0.897-0.909] | 0.852 [0.845-0.858] | 0.938 [0.937-0.939] | 0.893 [0.892-0.894] |
| LightGBM | 0.940 [0.937-0.942] | 0.885 [0.881-0.888] | 0.884 [0.876-0.891] | 0.838 [0.831-0.845] | **0.966** [0.964-0.967] | 0.931 [0.930-0.932] |
| RandomForest | 0.936 [0.934-0.939] | 0.886 [0.883-0.890] | 0.895 [0.888-0.901] | 0.848 [0.841-0.854] | 0.955 [0.954-0.956] | 0.920 [0.919-0.921] |
| CatBoost | 0.942 [0.939-0.944] | 0.891 [0.888-0.895] | 0.907 [0.901-0.914] | 0.856 [0.849-0.862] | 0.961 [0.960-0.962] | 0.926 [0.925-0.927] |
| XGBoost | 0.943 [0.940-0.946] | 0.892 [0.889-0.895] | 0.907 [0.900-0.914] | 0.861 [0.854-0.867] | 0.965 [0.964-0.966] | 0.931 [0.929-0.932] |
| TabPFN-$k$NN | 0.943 [0.941-0.946] | 0.891 [0.887-0.894] | 0.922 [0.916-0.927] | 0.864 [0.857-0.871] | 0.955 [0.953-0.956] | 0.916 [0.915-0.917] |
| LoCalPFN | **0.958** [0.956-0.960] | **0.908** [0.905-0.911] | **0.937** [0.931-0.942] | **0.882** [0.875-0.889] | **0.968** [0.967-0.969] | **0.934** [0.933-0.935] |

Table 1: AUC scores and confidence intervals for all 95 datasets, 47 small datasets, and 48 medium/large datasets.

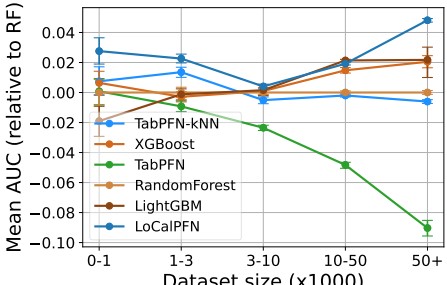

Figure 5: AUC vs. Size

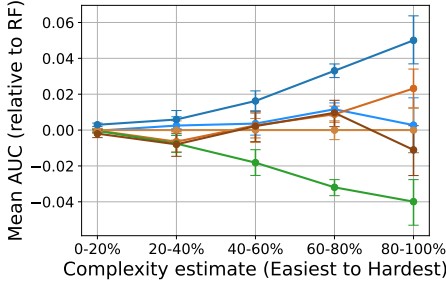

Figure 6: AUC vs. Complexity

Figure 7 (left) and Table 8, we see that this indeed improves performance over TabPFN. However, applying TabPFN-$k$NN on top of a naïvely fine-tuned model does not improve performance further. Therefore, it is crucial to fine-tune the model end-to-end *with* the retrieval (LocalPFN).

**Choice of Embedding** We also try different embeddings. The simplest approach is to use the raw standardized features for the nearest neighbour retrieval. In Figure 7 (centre) and Table 9, it is shown that this simple approach is actually very competitive. We compare it to two additional approaches: using one hot encodings (when the size of the resulting vector does not exceed 100 features), and the output of the encoder layer of TabPFN. For the latter, we recompute the search index every 30 gradient steps. The results show that the former, using one hot encodings, does lead to some improvement, however mostly for smaller datasets (see Figure 11 and Figure 12).

**Why do simple embeddings work so well?** While Tabular data is complex in many regards (Grinsztajn et al., 2022), features in tabular data are often semantically meaningful. For this reason, we expect distances that decompose over individual features (i.e $d(x, x') = \sum_i d_i(x_i, x'_i)$) to be a good inductive bias for tabular data, especially when it is normalized. This would not be the case for most natural signals.

**Importance of Using a Local Context** Up until now, we have mostly compared to TabPFN with a random context of size 1,000. To prove our point that using a local context is inherently better than a global context (same context for all queries), we attempt to find the best model using a global context by first using an ensemble of 32 TabPFN models (with randomized feature and class ordering as in Hollmann et al. (2023)), which we denote TabPFN-32ens, and then by increasing the context size of the ensembled TabPFN to 3,000 (TabPFN-3k-32ens). As depicted in Figure 7 (right) and detailed in Table 10, while improving significantly upon TabPFN, these are still not competitive even with our TabPFN-$k$NN. As one can criticize the use of a single context to classify queries, we further experimented with a "Bayesian" view of the probability by averaging it over contexts $p_\theta(y_{qy} \mid x_{qy}, \mathcal{D}_{train}) \triangleq \int_\mathcal{C} p_\theta(y_{qy} \mid x_{qy}, \mathcal{C})p(\mathcal{C} \mid \mathcal{D}_{train})d\mathcal{C}$, where $\mathcal{C}$ is a context obtained from the training data $\mathcal{D}_{train}$. We experimented with splitting $\mathcal{D}_{train}$ into chunks of size 3,000, and averaging the probabilities over those chunks. We call this method TabPFN-32ens-3k-int (for *integral*) and show that, while it does improve upon the single random context, it does not outperform TabPFN-$k$NN. Additionally, this method is very expensive as: (i) using 3,000 context examples is GPU memory intensive, and (ii) the integral over chunks makes the inference scale as $\mathcal{O}(N)$. The last method we compared to is "In-Context Distillation" (Ma et al., 2024) (TabPFN-ICD) where, similarly to Feuer et al. (2024), the authors directly optimize the context. While this last method leads to better performance (including on larger datasets, see Figure 12), since it performs task-specific tuning it is more comparable to LoCalPFN, which remains superior.

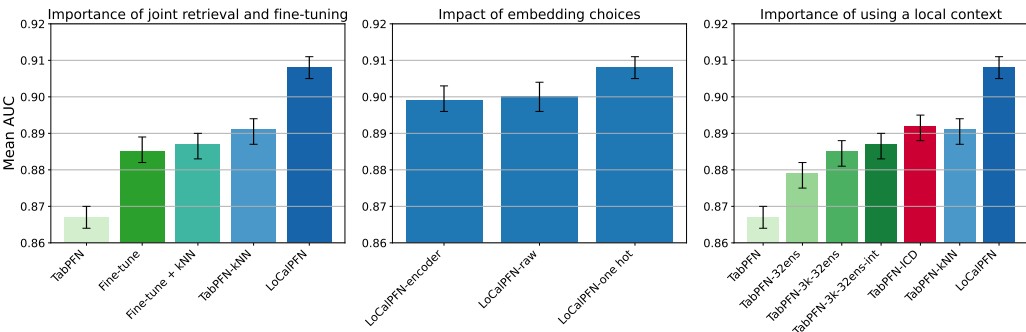

Figure 7: Ablations for different design choices on all 95 datasets. **Left:** Fine-tuning jointly with retrieval yields better performance. **Centre:** The choice of embeddings for retrieval does not change the performance drastically but can lead to some improvements. **Right:** Methods using a context that does not depend on the current query do not match the performance of methods that use a local context.

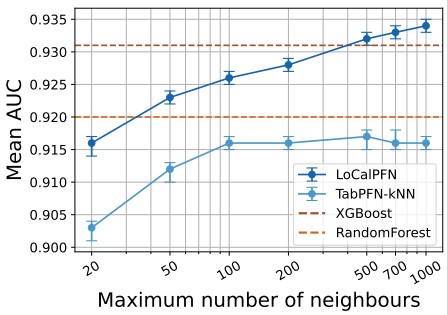

Figure 8: Ablating max # of neighbours

**Sensitivity to Number of Neighbours**  We also ablate the choice of the number of neighbours used as context. This is the only hyperparameter for TabPFN-$k$NN and also an important hyperparameter for LoCalPFN. In practice, for the number of neighbours, we use the minimum of (i) 10 times the square root of the training set size, and (ii) a pre-defined maximum. For large datasets, the number of neighbours should roughly align with the pre-defined maximum. In Figure 8, we vary this pre-defined maximum while observing the mean AUC on the 48 medium/large datasets. We found that TabPFN-$k$NN is not very sensitive to this choice as long as it is at least 100. We also see that LoCalPFN is able to improve TabPFN-$k$NN on all context sizes. Surprisingly, we observe that LoCalPFN is able to outperform the random forest baseline using a maximum context size of only 50, and also outperform the XGBoost baseline with maximum context size of 500. The details of the ablation can be found in Table 11.

**Other Ablations**  We also ascertain the quality of the approximate local context in real datasets in Appendix A.5.5 and Table 12, and we provide a runtime analysis of various methods in Appendix A.5.6 and Figure 13.

## 5. Conclusion and Limitations

In this paper, we demonstrate how to use retrieval and fine-tuning to improve performance on tabular data classification tasks by introducing LoCalPFN as a version of this framework that uses TabPFN as the base model. LoCalPFN breaks new ground for neural approaches on tabular data, even showing improvements over workhorse tree-based techniques. We also provide TabPFN-$k$NN as a variant without fine-tuning, demonstrating its superiority over the base model and practical utility.

However, despite its successes, our framework also has some limitations. The first is that we have only shown that retrieval and fine-tuning improve TabPFN, since it is the only proven ICL-based tabular model. Thus, we cannot be certain that our ideas would directly transfer to new base models, although the success of these concepts in other domains provides some evidence. It is also worth noting that the original RAG paper (Lewis et al., 2020) only initially demonstrated success on BART. Next, the reliance on TabPFN as a base model brings some limitations: besides the constraints on number of features and classes discussed in Section 4.1, we are also unable to easily test our ideas in regression tasks since TabPFN is not designed for them. Although we expect these constraints to gradually be lifted as tabular foundation models improve and increase their scope, we also note that tree-based methods are not nearly as susceptible to these issues. Going further on the comparison with tree-based methods, while we note that LoCalPFN performs better than them in our experimental study, we also point out in Appendix A.5.6 that the runtime of LoCalPFN is slower. Yet it is still faster than other deep learning approaches, and the cheaper TabPFN-$k$NN variant runs as fast as any tree-based method on datasets we studied, while still attaining respectable performance. Overall, we believe that the benefits of our framework far outweigh the limitations, as LoCalPFN greatly expands the capabilities of deep learning on tabular data.

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

# A. Appendix

## A.1. Datasets

Table 2: 47 Small Datasets

| dataset | did | # instances | # feat | # classes | # cat | imbalance ratio |
|---|---|---|---|---|---|---|
| Australian | 146818 | 690 | 14 | 2 | 8 | 1.248 |
| LED-display-domain-7digit | 125921 | 500 | 7 | 10 | 0 | 1.541 |
| acute-inflammations | 10089 | 120 | 6 | 2 | 5 | 1.400 |
| balance-scale | 11 | 625 | 4 | 3 | 0 | 5.878 |
| banknote-authentication | 10093 | 1372 | 4 | 2 | 0 | 1.249 |
| blood-transfusion-service-center | 10101 | 748 | 4 | 2 | 0 | 3.202 |
| breast-cancer | 145799 | 286 | 9 | 2 | 9 | 2.365 |
| car-evaluation | 146192 | 1728 | 21 | 4 | 21 | 18.615 |
| car | 146821 | 1728 | 6 | 4 | 6 | 18.615 |
| climate-model-simulation-crashes | 146819 | 540 | 18 | 2 | 0 | 10.739 |
| cmc | 23 | 1473 | 9 | 3 | 7 | 1.889 |
| credit-g | 31 | 1000 | 20 | 2 | 13 | 2.333 |
| diabetes | 37 | 768 | 8 | 2 | 0 | 1.866 |
| dresses-sales | 125920 | 500 | 12 | 2 | 11 | 1.381 |
| fertility | 9984 | 100 | 9 | 2 | 0 | 7.333 |
| hayes-roth | 146063 | 160 | 4 | 3 | 0 | 2.097 |
| hill-valley | 145847 | 1212 | 100 | 2 | 0 | 1.000 |
| ilpd | 9971 | 583 | 10 | 2 | 1 | 2.491 |
| ionosphere | 145984 | 351 | 34 | 2 | 0 | 1.786 |
| iris | 59 | 150 | 4 | 3 | 0 | 1.000 |
| kc2 | 3913 | 522 | 21 | 2 | 0 | 3.879 |
| monks-problems-2 | 146065 | 601 | 6 | 2 | 6 | 1.917 |
| pc1 | 3918 | 1109 | 21 | 2 | 0 | 13.403 |
| pc3 | 3903 | 1563 | 37 | 2 | 0 | 8.769 |
| pc4 | 3902 | 1458 | 37 | 2 | 0 | 7.191 |
| postoperative-patient-data | 146210 | 88 | 8 | 2 | 8 | 2.667 |
| profb | 3561 | 672 | 9 | 2 | 4 | 2.000 |
| qsar-biodeg | 9957 | 1055 | 41 | 2 | 0 | 1.963 |
| socmob | 3797 | 1156 | 5 | 2 | 4 | 3.516 |
| sonar | 39 | 208 | 60 | 2 | 0 | 1.144 |
| steel-plates-fault | 146817 | 1941 | 27 | 7 | 0 | 12.236 |
| tae | 47 | 151 | 5 | 3 | 2 | 1.061 |
| tic-tac-toe | 49 | 958 | 9 | 2 | 9 | 1.886 |
| transplant | 3748 | 131 | 3 | 2 | 0 | 1.729 |
| vehicle | 53 | 846 | 18 | 4 | 0 | 1.095 |
| wdbc | 9946 | 569 | 30 | 2 | 0 | 1.684 |
| yeast | 145793 | 1269 | 8 | 4 | 0 | 2.704 |

Table 3: 48 Medium/Large Datasets

| dataset | did | # instances | # feat | # classes | # cat | imbalance ratio |
|---|---|---|---|---|---|---|
| GesturePhaseSegmentationProcessed | 14969 | 9873 | 32 | 5 | 0 | 2.956 |
| JapaneseVowels | 3510 | 9961 | 14 | 9 | 0 | 2.064 |
| MagicTelescope | 3954 | 19020 | 10 | 2 | 0 | 1.844 |
| MiniBooNE | 168335 | 130064 | 50 | 2 | 0 | 2.563 |
| PhishingWebsites | 14952 | 11055 | 30 | 2 | 30 | 1.257 |
| Satellite | 167211 | 5100 | 36 | 2 | 0 | 67.000 |
| adult-census | 3953 | 32561 | 14 | 2 | 8 | 3.153 |
| adult | 7592 | 48842 | 14 | 2 | 8 | 3.179 |
| artificial-characters | 14964 | 10218 | 7 | 10 | 0 | 2.360 |
| bank-marketing | 14965 | 45211 | 16 | 2 | 9 | 7.548 |
| cardiotocography | 9979 | 2126 | 35 | 10 | 0 | 10.925 |
| churn | 167141 | 5000 | 20 | 2 | 4 | 6.072 |
| connect-4 | 146195 | 67557 | 42 | 3 | 42 | 6.896 |
| eeg-eye-state | 14951 | 14980 | 14 | 2 | 0 | 1.228 |
| electricity | 219 | 45312 | 8 | 2 | 1 | 1.355 |
| elevators | 3711 | 16599 | 18 | 2 | 0 | 2.236 |
| first-order-theorem-proving | 9985 | 6118 | 51 | 6 | 0 | 5.255 |
| jannis | 168330 | 83733 | 54 | 4 | 0 | 22.835 |
| kc1 | 3917 | 2109 | 21 | 2 | 0 | 5.469 |
| kr-vs-kp | 3 | 3196 | 36 | 2 | 36 | 1.093 |
| magic | 146206 | 19020 | 10 | 2 | 0 | 1.844 |
| mfeat-fourier | 14 | 2000 | 76 | 10 | 0 | 1.000 |
| mfeat-karhunen | 16 | 2000 | 64 | 10 | 0 | 1.000 |
| mfeat-morphological | 18 | 2000 | 6 | 10 | 0 | 1.000 |

Table 3: 48 Medium/Large Datasets

| dataset | did | # instances | # feat | # classes | # cat | imblance ratio |
|---|---|---|---|---|---|---|
| mfeat-zernike | 22 | 2000 | 47 | 10 | 0 | 1.000 |
| mushroom | 24 | 8124 | 22 | 2 | 22 | 1.075 |
| numerai28.6 | 167120 | 96320 | 21 | 2 | 0 | 1.021 |
| nursery | 9892 | 12958 | 8 | 4 | 8 | 13.171 |
| optdigits | 28 | 5620 | 64 | 10 | 0 | 1.032 |
| ozone-level-8hr | 9978 | 2534 | 72 | 2 | 0 | 14.838 |
| page-blocks | 30 | 5473 | 10 | 5 | 0 | 175.464 |
| pendigits | 32 | 10992 | 16 | 10 | 0 | 1.084 |
| phoneme | 9952 | 5404 | 5 | 2 | 0 | 2.407 |
| pollen | 3735 | 3848 | 5 | 2 | 0 | 1.000 |
| satimage | 2074 | 6430 | 36 | 6 | 0 | 2.450 |
| segment | 146822 | 2310 | 16 | 7 | 0 | 1.000 |
| shuttle | 146212 | 58000 | 9 | 7 | 0 | 4558.600 |
| spambase | 43 | 4601 | 57 | 2 | 0 | 1.538 |
| splice | 45 | 3190 | 60 | 3 | 60 | 2.158 |
| sylvine | 168912 | 5124 | 20 | 2 | 0 | 1.000 |
| wall-robot-navigation | 9960 | 5456 | 24 | 4 | 0 | 6.723 |
| wilt | 146820 | 4839 | 5 | 2 | 0 | 17.540 |

Table 4: 71 Datasets Selected for Benchmarking Deep Learning Models

| dataset | did | # instances | # feat | # classes | # cat | imblance ratio |
|---|---|---|---|---|---|---|
| Australian | 146818 | 690 | 14 | 2 | 8 | 1.248 |
| LED-display-domain-7digit | 125921 | 500 | 7 | 10 | 0 | 1.541 |
| Satellite | 167211 | 5100 | 36 | 2 | 0 | 67.000 |
| acute-inflammations | 10089 | 120 | 6 | 2 | 5 | 1.400 |
| balance-scale | 11 | 625 | 4 | 3 | 0 | 5.878 |
| banknote-authentication | 10093 | 1372 | 4 | 2 | 0 | 1.249 |
| blood-transfusion-service-center | 10101 | 748 | 4 | 2 | 0 | 3.202 |
| breast-cancer | 145799 | 286 | 9 | 2 | 9 | 2.365 |
| car-evaluation | 146192 | 1728 | 21 | 4 | 21 | 18.615 |
| car | 146821 | 1728 | 6 | 4 | 6 | 18.615 |
| cardiotocography | 9979 | 2126 | 35 | 10 | 0 | 10.925 |
| churn | 167141 | 5000 | 20 | 2 | 4 | 6.072 |
| climate-model-simulation-crashes | 146819 | 540 | 18 | 2 | 0 | 10.739 |
| cmc | 23 | 1473 | 9 | 3 | 7 | 1.889 |
| credit-g | 31 | 1000 | 20 | 2 | 13 | 2.333 |
| diabetes | 37 | 768 | 8 | 2 | 0 | 1.866 |
| dresses-sales | 125920 | 500 | 12 | 2 | 11 | 1.381 |
| eeg-eye-state | 14951 | 14980 | 14 | 2 | 0 | 1.228 |
| fertility | 9984 | 100 | 9 | 2 | 0 | 7.333 |
| first-order-theorem-proving | 9985 | 6118 | 51 | 6 | 0 | 5.255 |
| hayes-roth | 146063 | 160 | 4 | 3 | 0 | 2.097 |
| hill-valley | 145847 | 1212 | 100 | 2 | 0 | 1.000 |
| ilpd | 9971 | 583 | 10 | 2 | 1 | 2.491 |
| ionosphere | 145984 | 351 | 34 | 2 | 0 | 1.786 |
| iris | 59 | 150 | 4 | 3 | 0 | 1.000 |
| kc1 | 3917 | 2109 | 21 | 2 | 0 | 5.469 |
| kc2 | 3913 | 522 | 21 | 2 | 0 | 3.879 |
| kr-vs-kp | 3 | 3196 | 36 | 2 | 36 | 1.093 |
| mfeat-fourier | 14 | 2000 | 76 | 10 | 0 | 1.000 |
| mfeat-karhunen | 16 | 2000 | 64 | 10 | 0 | 1.000 |
| mfeat-morphological | 18 | 2000 | 6 | 10 | 0 | 1.000 |
| mfeat-zernike | 22 | 2000 | 47 | 10 | 0 | 1.000 |
| monks-problems-2 | 146065 | 601 | 6 | 2 | 6 | 1.917 |
| mushroom | 24 | 8124 | 22 | 2 | 22 | 1.075 |
| optdigits | 28 | 5620 | 64 | 10 | 0 | 1.032 |
| ozone-level-8hr | 9978 | 2534 | 72 | 2 | 0 | 14.838 |
| page-blocks | 30 | 5473 | 10 | 5 | 0 | 175.464 |
| pc1 | 3918 | 1109 | 21 | 2 | 0 | 13.403 |
| pc3 | 3903 | 1563 | 37 | 2 | 0 | 8.769 |
| pc4 | 3902 | 1458 | 37 | 2 | 0 | 7.191 |
| phoneme | 9952 | 5404 | 5 | 2 | 0 | 2.407 |
| pollen | 3735 | 3848 | 5 | 2 | 0 | 1.000 |
| postoperative-patient-data | 146210 | 88 | 8 | 2 | 8 | 2.667 |
| profb | 3561 | 672 | 9 | 2 | 4 | 2.000 |
| qsar-biodeg | 9957 | 1055 | 41 | 2 | 0 | 1.963 |
| satimage | 2074 | 6430 | 36 | 6 | 0 | 2.450 |
| segment | 146822 | 2310 | 16 | 7 | 0 | 1.000 |
| socmob | 3797 | 1156 | 5 | 2 | 4 | 3.516 |

Table 4: 71 Datasets Selected for Benchmarking Deep Learning Models

| dataset | did | # instances | # feat | # classes | # cat | imblance ratio |
|---------|-----|-------------|--------|-----------|-------|----------------|
| sonar | 39 | 208 | 60 | 2 | 0 | 1.144 |
| spambase | 43 | 4601 | 57 | 2 | 0 | 1.538 |
| splice | 45 | 3190 | 60 | 3 | 60 | 2.158 |
| steel-plates-fault | 146817 | 1941 | 27 | 7 | 0 | 12.236 |
| tae | 47 | 151 | 5 | 3 | 2 | 1.061 |
| tic-tac-toe | 49 | 958 | 9 | 2 | 9 | 1.886 |
| transplant | 3748 | 131 | 3 | 2 | 0 | 1.729 |
| vehicle | 53 | 846 | 18 | 4 | 0 | 1.095 |
| wall-robot-navigation | 9960 | 5456 | 24 | 4 | 0 | 6.723 |
| wdbc | 9946 | 569 | 30 | 2 | 0 | 1.684 |
| wilt | 146820 | 4839 | 5 | 2 | 0 | 17.540 |
| yeast | 145793 | 1269 | 8 | 4 | 0 | 2.704 |

## A.2. Experiment Details

### A.2.1. BASELINE DETAILS

We use the experimental results from TabZilla (McElfresh et al., 2023) when they are available. These results include the tree-based models and the deep learning model baselines. These results can be found in `https://github.com/naszilla/tabzilla` and `https://drive.google.com/drive/folders/1cHisTmruPHDCYVOYnaqvTdybLngMkB8R`.

For different variations of TabPFN inference techniques, we conduct experiments directly using the TabPFN repository `https://github.com/automl/TabPFN`.

### A.2.2. LoCalPFN DETAILS

For all TabPFN-$k$NN experiments, we use a fixed number of neighbours equal to the minimum of (i) 10 times the square root of the dataset size, and (ii) 1000. We find this works well in practice since it adapts to small and large datasets. In practice, we use a batch size of 512 for inference using the faiss library for speedup.

For LoCalPFN experiments, we use the exact same setup as TabPFN-$k$NN during inference. Therefore, at step 0, LoCalPFN and TabPFN-$k$NN are equivalent. For training LoCalPFN, we adopt the AdamW (Loshchilov & Hutter, 2019) optimizer with a learning rate of 0.01 and weight decay of 0.01. We do not have warmup or a learning rate scheduler. For the approximate local context for training, we use the same number of neighbours as TabPFN-$k$NN. We use a fixed number of query points (1,000) sampled from the training set and a batch of 2. For our reported results, we also use one-hot encoding for neighbour retrieval and inference. In addition, we evaluate our model every 30 gradient steps and apply early stopping based on the validation set AUC for each fold respectively.

All experiments for our proposed methods can be run on a machine with a single NVIDIA RTX 6000 GPU Ada Generation, 995Gi RAM, and AMD Ryzen Threadripper PRO 5995WX 64-Cores CPU. Additional runtime analysis can be found in Figure 13.

## A.3. Additional Experiments

### A.3.1. COMPARISON TO DEEP LEARNING MODELS

In addition to tree-based models, we also compare LoCalPFN and TabPFN-$k$NN with deep learning based methods. We use the results directly from the TabZilla repository. However, due to the fact that a lot of the deep learning baselines are very computationally expensive, many of them were not able to run on all datasets. Therefore, we propose a subset of the 95 datasets which contains 71 datasets upon which all the deep learning methods could run. The details of the 71 dataset subset can be found in Table 4.

The complete results can be found in Table 5. We can see that LoCalPFN still outperforms all other models.

### A.3.2. COMPARISON WITH OTHER METRICS

Here we also compare the performance of LoCalPFN with other models using accuracy and F1 score as the metric. We can observe a similar pattern here: LoCalPFN either matches or outperforms other models on either of these metrics as well.

Table 5: LoCalPFN outperforms deep learning baselines significantly.

| Algorithm | All 71 Datasets | |
|---|---|---|
| | IQM AUC | Mean AUC |
| VIME | 0.771 [0.760-0.782] | 0.741 [0.732-0.750] |
| rtdl_MLP | 0.855 [0.848-0.862] | 0.812 [0.806-0.818] |
| TabNet | 0.881 [0.874-0.888] | 0.825 [0.818-0.832] |
| STG | 0.877 [0.872-0.883] | 0.829 [0.823-0.834] |
| rtdl_ResNet | 0.917 [0.912-0.922] | 0.862 [0.857-0.867] |
| rtdl_FTTransformer | 0.919 [0.913-0.924] | 0.869 [0.864-0.874] |
| TabPFN | 0.929 [0.925-0.932] | 0.875 [0.871-0.879] |
| Fine-Tune | 0.936 [0.932-0.939] | 0.881 [0.876-0.886] |
| TabPFN-$k$NN | 0.948 [0.944-0.951] | 0.889 [0.884-0.894] |
| LoCalPFN-encoder | **0.956** [0.953-0.959] | 0.892 [0.887-0.897] |
| LoCalPFN-raw | **0.957** [0.954-0.960] | 0.893 [0.887-0.898] |
| Fine-Tune+$k$NN | 0.951 [0.948-0.954] | 0.893 [0.888-0.897] |
| LoCalPFN | **0.959** [0.956-0.962] | **0.903** [0.899-0.907] |

Table 6: Accuracy comparison for LoCalPFN and the baseline models.

| Algorithm | All | | Small | | Medium/Large | |
|---|---|---|---|---|---|---|
| | IQM Acc | Mean Acc | IQM Acc | Mean Acc | IQM Acc | Mean Acc |
| TabPFN | 0.856 [0.853-0.859] | 0.817 [0.815-0.820] | 0.836 [0.830-0.842] | 0.806 [0.801-0.811] | 0.871 [0.869-0.872] | 0.828 [0.826-0.830] |
| TabPFN 3k | 0.862 [0.859-0.865] | 0.823 [0.820-0.826] | 0.839 [0.833-0.845] | 0.808 [0.803-0.813] | 0.881 [0.879-0.882] | 0.837 [0.835-0.839] |
| RandomForest | 0.875 [0.873-0.878] | 0.839 [0.837-0.841] | 0.834 [0.827-0.840] | 0.807 [0.802-0.812] | 0.900 [0.899-0.901] | 0.866 [0.865-0.867] |
| LightGBM | 0.878 [0.875-0.881] | 0.842 [0.839-0.845] | 0.830 [0.824-0.837] | 0.807 [0.802-0.812] | **0.918** [0.916-0.919] | 0.886 [0.885-0.887] |
| CatBoost | 0.883 [0.880-0.886] | 0.847 [0.844-0.849] | 0.844 [0.838-0.850] | 0.815 [0.810-0.820] | 0.908 [0.907-0.909] | 0.876 [0.875-0.877] |
| XGBoost | 0.889 [0.886-0.892] | 0.848 [0.845-0.851] | 0.840 [0.833-0.847] | 0.811 [0.804-0.817] | **0.919** [0.918-0.920] | 0.887 [0.886-0.888] |
| TabPFN-$k$NN | 0.877 [0.874-0.879] | 0.843 [0.841-0.845] | 0.856 [0.851-0.862] | 0.825 [0.820-0.829] | 0.891 [0.890-0.892] | 0.861 [0.860-0.862] |
| LoCalPFN | **0.902** [0.900-0.905] | **0.865** [0.863-0.868] | **0.875** [0.869-0.881] | **0.840** [0.835-0.845] | 0.918 [0.916-0.919] | **0.890** [0.889-0.891] |

Table 7: F1 score comparison for LoCalPFN and the baseline models.

| Algorithm | All | | Small | | Medium/Large | |
|---|---|---|---|---|---|---|
| | IQM F1 | Mean F1 | IQM F1 | Mean F1 | IQM F1 | Mean F1 |
| TabPFN | 0.843 [0.840-0.846] | 0.796 [0.794-0.799] | 0.818 [0.812-0.825] | 0.783 [0.778-0.789] | 0.861 [0.859-0.863] | 0.809 [0.807-0.811] |
| TabPFN 3k | 0.850 [0.847-0.853] | 0.801 [0.798-0.804] | 0.821 [0.814-0.828] | 0.784 [0.779-0.789] | 0.872 [0.870-0.874] | 0.818 [0.816-0.820] |
| RandomForest | 0.875 [0.872-0.877] | 0.837 [0.835-0.839] | 0.831 [0.824-0.838] | 0.805 [0.800-0.811] | 0.900 [0.898-0.901] | 0.863 [0.862-0.864] |
| LightGBM | 0.877 [0.874-0.881] | 0.841 [0.838-0.844] | 0.829 [0.823-0.836] | 0.806 [0.801-0.811] | **0.917** [0.916-0.919] | **0.885** [0.884-0.886] |
| CatBoost | 0.882 [0.879-0.885] | 0.845 [0.843-0.848] | 0.842 [0.836-0.849] | 0.814 [0.808-0.819] | 0.908 [0.907-0.909] | 0.875 [0.874-0.876] |
| XGBoost | 0.888 [0.885-0.891] | 0.847 [0.844-0.850] | 0.839 [0.832-0.846] | 0.810 [0.804-0.816] | **0.919** [0.918-0.920] | **0.886** [0.885-0.887] |
| TabPFN-$k$NN | 0.867 [0.864-0.870] | 0.829 [0.827-0.832] | 0.841 [0.834-0.847] | 0.804 [0.800-0.809] | 0.884 [0.883-0.886] | 0.854 [0.853-0.855] |
| LoCalPFN | **0.897** [0.894-0.899] | **0.859** [0.856-0.861] | **0.869** [0.863-0.874] | **0.832** [0.827-0.837] | 0.915 [0.913-0.916] | **0.885** [0.884-0.886] |

## A.4. Additional Analyses

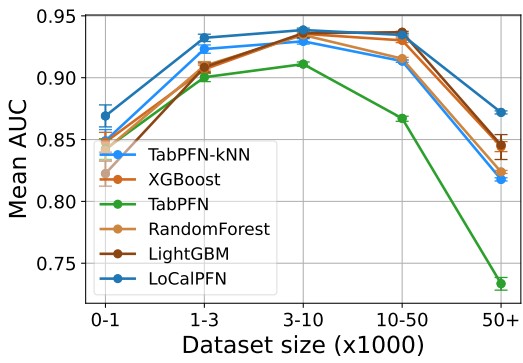
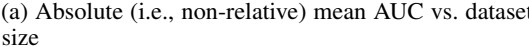

(a) Absolute (i.e., non-relative) mean AUC vs. dataset size

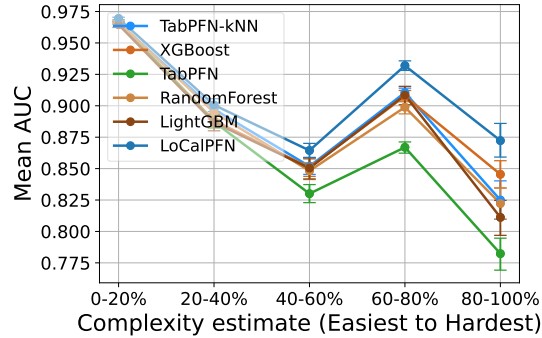

(b) Absolute (i.e., non-relative) mean AUC vs. complexity

Figure 9: Analysis of AUC as a function of size and complexity. TabPFN fails to scale both in size and complexity while LoCalPFN is able to still outperform on the far end of the spectrum.

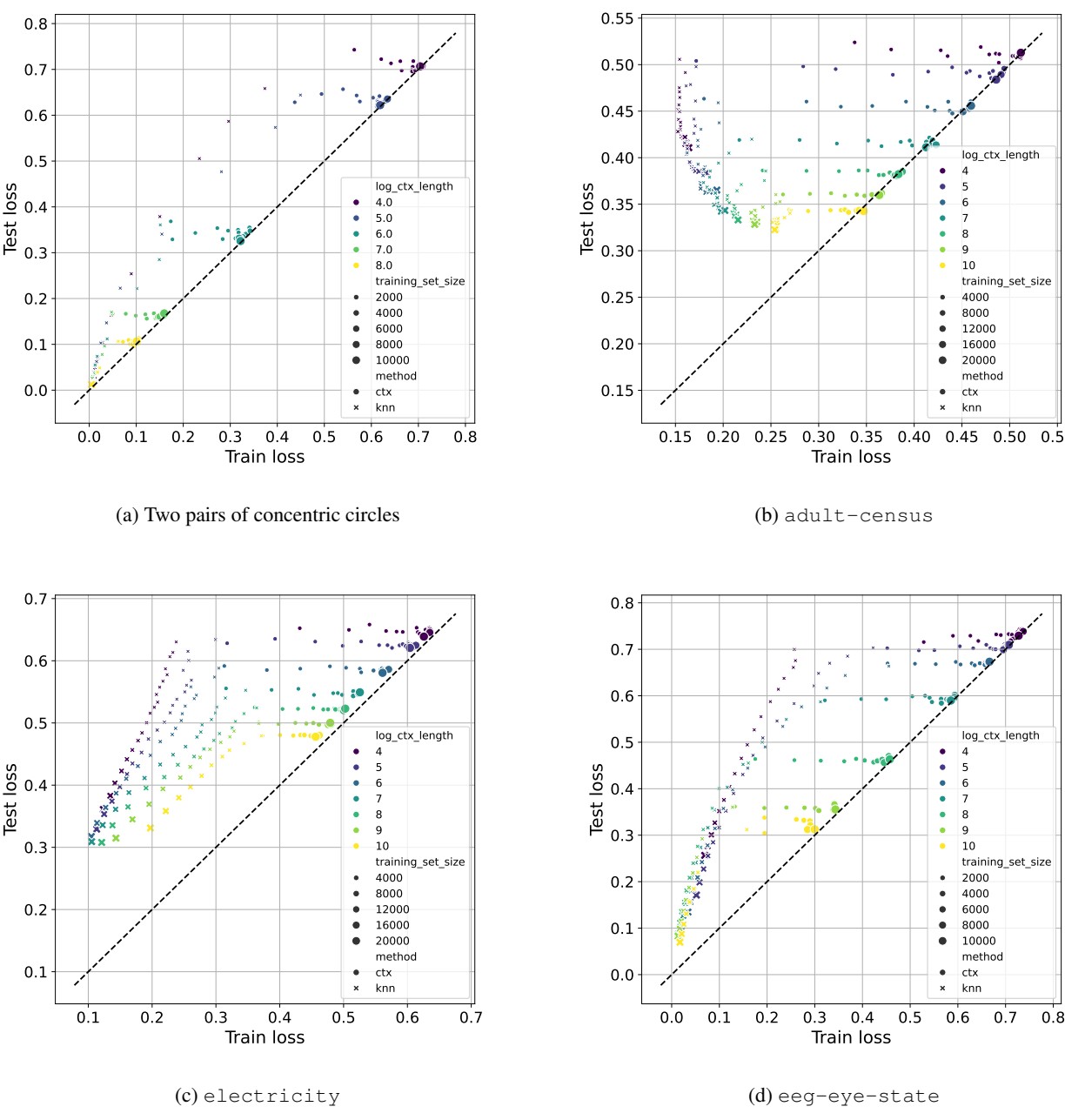

(a) Two pairs of concentric circles

(b) `adult-census`

(c) `electricity`

(d) `eeg-eye-state`

Figure 10: Test loss vs. training loss for TabPFN-$k$NN (crosses), TabPFN (circles) for different dataset sizes and context/number of neighbours used on four datasets. We observe generally that for low number of neighbours (dark crosses) and especially for small datasets (small crosses) there is significant overfitting (higher test loss than train loss). TabPFN tends to overfit less, especially on larger datasets, which is expected. Overall, using TapPFN-$k$NN results in better underfitting/overfitting trade-offs where we obtain both lower test and train losses, however the gap between them increases.

## A.5. Ablation Studies

Figure 11 and Figure 12 show summaries of ablations on only the small datasets, and only the medium/large datasets, respectively. In the remainder of this subsection we see tables that show even further detail on the results presented in the main text.

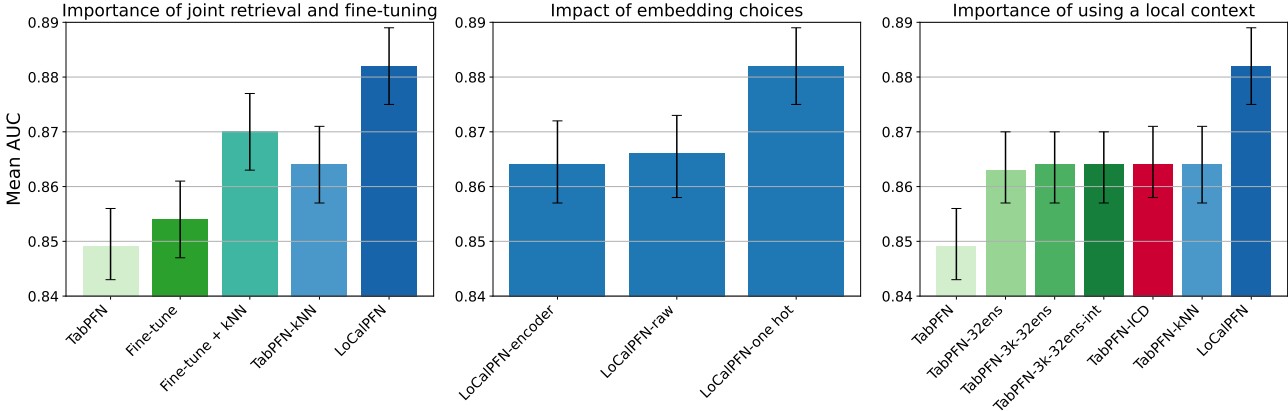

Figure 11: Ablations on Small Datasets

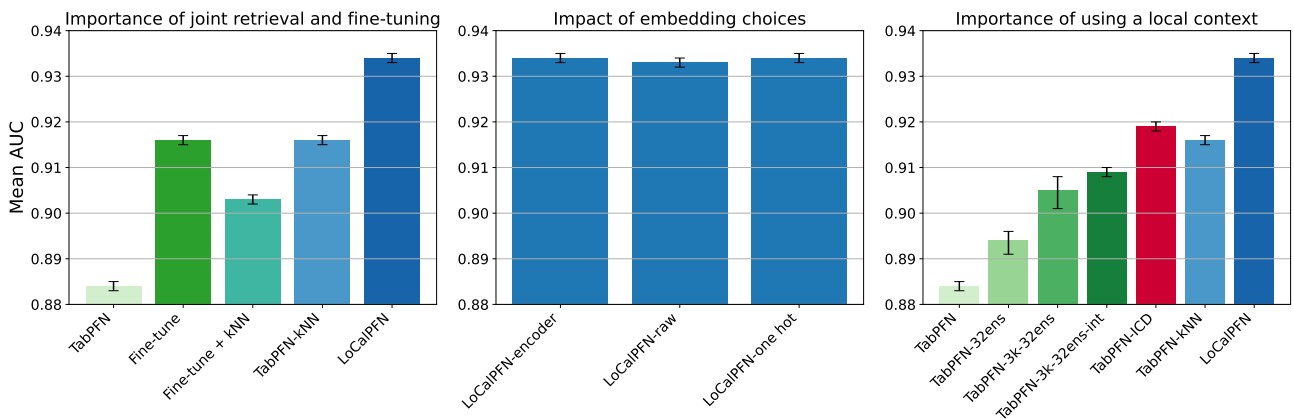

Figure 12: Ablations on Medium/Large Datasets

A.5.1. IMPORTANCE OF JOINT RETRIEVAL AND FINE-TUNING

Table 8: Ablation for fine-tuning. Applying TabPFN-$k$NN on a fine-tuned model degrades the overall performance. On the other hand, performing local calibration by jointly retrieving and fine-tuning improve performance drastically.

| | All | | Small | | Medium/Large | |
|---|---|---|---|---|---|---|
| Algorithm | IQM AUC | Mean AUC | IQM AUC | Mean AUC | IQM AUC | Mean AUC |
| TabPFN | 0.917 [0.914-0.919] | 0.867 [0.864-0.870] | 0.898 [0.892-0.904] | 0.849 [0.843-0.856] | 0.927 [0.925-0.929] | 0.884 [0.883-0.885] |
| Fine-Tune | 0.934 [0.932-0.937] | 0.885 [0.882-0.889] | 0.905 [0.897-0.911] | 0.854 [0.847-0.861] | 0.953 [0.951-0.954] | 0.916 [0.915-0.917] |
| Fine-Tune + $k$NN | 0.938 [0.935-0.940] | 0.887 [0.883-0.890] | **0.928** [0.922-0.933] | **0.870** [0.863-0.877] | 0.946 [0.945-0.948] | 0.903 [0.902-0.904] |
| TabPFN-$k$NN | 0.943 [0.941-0.946] | 0.891 [0.887-0.894] | 0.922 [0.916-0.927] | 0.864 [0.857-0.871] | 0.955 [0.953-0.956] | 0.916 [0.915-0.917] |
| LoCalPFN | **0.958** [0.956-0.960] | **0.908** [0.905-0.911] | **0.937** [0.931-0.942] | **0.882** [0.875-0.889] | **0.968** [0.967-0.969] | **0.934** [0.933-0.935] |

## A.5.2. CHOICE OF FEATURE ENCODING

Table 9: Ablation for choices of embedding. Converting categorical variables to one-hot gives a relatively moderate gain over other configurations.

| | All | | Small | | Medium/Large | |
|---|---|---|---|---|---|---|
| Algorithm | IQM AUC | Mean AUC | IQM AUC | Mean AUC | IQM AUC | Mean AUC |
| LoCalPFN-encoder | 0.955 [0.953-0.957] | 0.899 [0.896-0.903] | 0.926 [0.920-0.932] | 0.864 [0.857-0.872] | 0.969 [0.967-0.969] | 0.934 [0.933-0.935] |
| LoCalPFN-raw | 0.956 [0.954-0.958] | 0.900 [0.896-0.904] | 0.928 [0.922-0.934] | 0.866 [0.858-0.873] | 0.968 [0.967-0.969] | 0.933 [0.932-0.934] |
| LoCalPFN-one_hot | 0.958 [0.956-0.960] | 0.908 [0.905-0.911] | 0.937 [0.931-0.942] | 0.882 [0.875-0.889] | 0.968 [0.967-0.969] | 0.934 [0.933-0.935] |

## A.5.3. OTHER INFERENCE METHODS OF TABPFN

Table 10 shows the detailed performance values for TabPFN with different inference methods.

Table 10: Ablation for different TabPFN inference methods.

| | All | | Small | | Medium/Large | |
|---|---|---|---|---|---|---|
| Algorithm | IQM AUC | Mean AUC | IQM AUC | Mean AUC | IQM AUC | Mean AUC |
| TabPFN-1k-1ens | 0.917 [0.914-0.919] | 0.867 [0.864-0.870] | 0.898 [0.892-0.904] | 0.849 [0.843-0.856] | 0.927 [0.926-0.929] | 0.884 [0.883-0.885] |
| TabPFN-1k-32ens | 0.936 [0.934-0.938] | 0.879 [0.875-0.882] | 0.923 [0.917-0.929] | 0.863 [0.857-0.870] | 0.943 [0.941-0.944] | 0.894 [0.891-0.896] |
| TabPFN-3k-32ens | 0.943 [0.941-0.945] | 0.885 [0.881-0.888] | 0.924 [0.918-0.930] | 0.864 [0.857-0.870] | 0.954 [0.953-0.955] | 0.905 [0.901-0.908] |
| TabPFN-3k-32ens-int | 0.945 [0.942-0.947] | 0.887 [0.883-0.890] | 0.924 [0.918-0.930] | 0.864 [0.857-0.870] | 0.956 [0.955-0.957] | 0.909 [0.908-0.910] |
| TabPFN-ICD | 0.946 [0.944-0.948] | 0.892 [0.888-0.895] | 0.924 [0.919-0.930] | 0.864 [0.858-0.871] | 0.958 [0.957-0.959] | 0.919 [0.918-0.920] |
| TabPFN-$k$NN | 0.943 [0.941-0.946] | 0.891 [0.887-0.894] | 0.922 [0.916-0.928] | 0.864 [0.857-0.871] | 0.955 [0.953-0.956] | 0.916 [0.915-0.917] |
| LoCalPFN | **0.958** [0.956-0.960] | **0.908** [0.905-0.911] | **0.937** [0.931-0.942] | **0.882** [0.876-0.888] | **0.968** [0.967-0.969] | **0.934** [0.933-0.935] |

## A.5.4. ABLATION FOR MAXIMUM NUMBER OF NEIGHBOURS

Table 11 shows the detailed performance values for varying size of maximum number of neighbours.

Table 11: Ablation for sensitivity of $k$. The number after c indicates the maximum number of neighbours used.

| Algorithm | All | | Small | | Medium/Large | |
|---|---|---|---|---|---|---|
| | IQM AUC | Mean AUC | IQM AUC | Mean AUC | IQM AUC | Mean AUC |
| TabPFN-$k$NN-c20 | 0.923 [0.920-0.925] | 0.874 [0.871-0.878] | 0.894 [0.887-0.901] | 0.845 [0.838-0.852] | 0.937 [0.936-0.939] | 0.903 [0.901-0.904] |
| TabPFN-$k$NN-c50 | 0.935 [0.933-0.938] | 0.886 [0.882-0.889] | 0.911 [0.905-0.917] | 0.859 [0.852-0.866] | 0.949 [0.948-0.950] | 0.912 [0.910-0.913] |
| TabPFN-$k$NN-c100 | 0.943 [0.940-0.945] | 0.890 [0.887-0.894] | 0.921 [0.916-0.927] | 0.864 [0.857-0.871] | 0.954 [0.952-0.955] | 0.916 [0.915-0.917] |
| TabPFN-$k$NN-c200 | 0.943 [0.941-0.946] | 0.890 [0.887-0.894] | 0.922 [0.916-0.927] | 0.864 [0.857-0.871] | 0.955 [0.953-0.956] | 0.916 [0.915-0.917] |
| TabPFN-$k$NN-c500 | 0.943 [0.941-0.946] | 0.891 [0.887-0.894] | 0.922 [0.916-0.928] | 0.864 [0.857-0.871] | 0.955 [0.953-0.956] | 0.917 [0.915-0.918] |
| TabPFN-$k$NN-c700 | 0.943 [0.941-0.946] | 0.891 [0.887-0.894] | 0.922 [0.916-0.927] | 0.864 [0.857-0.871] | 0.955 [0.953-0.956] | 0.916 [0.915-0.918] |
| TabPFN-$k$NN-c1000 | 0.943 [0.941-0.946] | 0.891 [0.887-0.894] | 0.922 [0.916-0.927] | 0.864 [0.857-0.871] | 0.955 [0.953-0.956] | 0.916 [0.915-0.917] |
| LoCalPFN-c20 | 0.941 [0.938-0.944] | 0.890 [0.887-0.894] | 0.920 [0.913-0.926] | 0.865 [0.858-0.872] | 0.953 [0.952-0.955] | 0.916 [0.914-0.917] |
| LoCalPFN-c50 | 0.950 [0.948-0.953] | 0.898 [0.894-0.902] | 0.932 [0.925-0.938] | 0.873 [0.865-0.881] | 0.960 [0.959-0.961] | 0.923 [0.922-0.924] |
| LoCalPFN-c100 | 0.953 [0.951-0.955] | 0.901 [0.897-0.904] | 0.932 [0.926-0.938] | 0.875 [0.868-0.882] | 0.963 [0.962-0.964] | 0.926 [0.925-0.927] |
| LoCalPFN-c200 | 0.955 [0.953-0.958] | 0.904 [0.900-0.908] | 0.935 [0.929-0.941] | 0.879 [0.872-0.886] | 0.965 [0.964-0.966] | 0.928 [0.927-0.929] |
| LoCalPFN-c500 | 0.957 [0.955-0.959] | 0.905 [0.901-0.908] | 0.935 [0.930-0.941] | 0.877 [0.870-0.883] | 0.968 [0.967-0.968] | 0.932 [0.931-0.933] |
| LoCalPFN-c700 | 0.958 [0.955-0.960] | 0.906 [0.902-0.910] | 0.935 [0.930-0.941] | 0.879 [0.871-0.886] | 0.968 [0.967-0.969] | 0.933 [0.932-0.934] |
| LoCalPFN-c1000 | 0.958 [0.956-0.960] | 0.908 [0.905-0.911] | 0.937 [0.931-0.942] | 0.882 [0.875-0.889] | 0.968 [0.967-0.969] | 0.934 [0.933-0.935] |

### A.5.5. QUALITY OF EFFICIENT LOCAL CONTEXT

In order to show the efficacy of the efficient local context, we compare LoCalPFN with the exact version where we use the exact neighbours for the context during training. In Table 12, LoCalPFN-exact-b32 indicates the aforementioned configuration with a batch size of 32, which is capped because of the GPU memory constraint. We compare this with another variant of LoCalPFN where we use 32 queries for training, i.e., LoCalPFN-approx-q32. These two variants turn out to have very similar AUCs, which indicates the efficacy of the efficient approximate neighbour search method.

Table 12: Exact nearest neighbour search vs. approximate nearest neighbour search.

| Algorithm | Medium/Large | |
|---|---|---|
| | IQM AUC | Mean AUC |
| LoCalPFN-exact-b32 | 0.967 [0.966-0.968] | 0.931 [0.930-0.932] |
| LoCalPFN-approx-q32 | 0.968 [0.967-0.968] | 0.931 [0.930-0.932] |
| LoCalPFN | 0.968 [0.967-0.969] | 0.934 [0.933-0.935] |

### A.5.6. RUN TIME ANALYSIS

We also conduct a run time analysis for LoCalPFN, TabPFN-$k$NN, and other tree-based models. We decided to measure the time against the mean AUC on the test set. The best algorithm should take the least amount of time and achieve the highest AUC. Here we measure the time as the total time of training and evaluation. In Figure 13, we can see that the general trend for all algorithms shows a positive correlation between time and AUC. In particular, we can observe TabPFN-$k$NN runs surprisingly fast and also achieves quite high AUC, only very slightly below XGBoost. The fast run time together with very few hyperparameter suggests that this should be a very good model to be used in practical machine learning engineering and research.

We also see that LoCalPFN achieves significantly higher performance even though it suffers from higher run time. We also want to point out that other deep learning methods shown in Table 5 take an even longer time for training and evaluation.

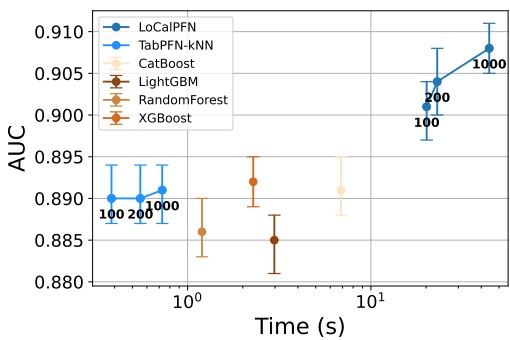

Figure 13: AUC vs Run Time for all 95 datasets. TabPFN-$k$NN has very low run time and comparable AUC to tree-based models while LoCalPFN is able to achieve the highest AUC overall. We use bold text to denote maximum number of neighbours $k$ used.

