# OpenReview forum: "Retrieval & Fine-Tuning for In-Context Tabular Models"
_ICML.cc/2024/Workshop/ICL — ICML 2024 Workshop ICL Poster_

### Official Review · Reviewer_nNno · 2024-06-04
**A well-written and novel paper with a minor flaw in the analysis of the results and minor unclarities in the experimental design**

**Rating:** 2
**Fit:** 3
**Confidence:** 3

**Workshop Review:**

# Contribution Summary
The paper proposes an approach to joint retrieval and fine-tuning for tabular in-context learning models to enhance performance and scalability.
To do so, they propose an efficient fine-tuning paradigm when fine-tuning w.r.t. kNN-based retrieval.
Their approach is built up on TabPFN but is, in theory, applicable to any in-context tabular model.
Their proposed method outperforms all compared baselines and other TabPFN variants.
Moreover, their ablations show the power of joint retrieval and fine-tuning.

# Recommendation Summary

This paper is a great fit for the ICL workshop. It is well-written and proposes a novel and likely impactful idea for getting the most out of tabular in-context learning models.

The biggest downside is a minor flaw related to how results are analyzed. Additionally, as far as I can tell, key information about their method's experimental design and usage related to fine-tuning data is missing. More on these points below.

Nevertheless, I highly recommend the acceptance of this work.

# Detailed Feedback

## Non-normalized Aggregation Across Diverse Tasks

In all parts of this work, the authors used a non-normalized aggregation of the AUC. The only alternative perspective for aggregation is using the IQM instead of the mean itself. I believe this choice of analyzing the results is extremely impactful for tabular data and the presented conclusion. Furthermore, unlike most prior work on tabular data or TabPFN, this work does not present average ranks.

Normalization before aggregation is common in tabular data [1, 2, 3] and very reasonable choices in the diverse landscape of machine learning tasks. Obtaining a higher metric score (be it ROC AUC or Accuracy) on one task does not mean it was "hard" to obtain this score for the given task, which makes the non-normalized average misguided; see [1] for an extended discussion. The work by McElfresh et al. (2023), which the authors use to get results and data for their baseline, also always used normalization or ranks.

The results in this work are likely not entirely wrong, but placing the result into context is very hard. Moreover, confounding factors such as simply getting better on a few (large) datasets might be the reasons for a higher non-normalized average instead of having a better method.

To me, this also explains the supposedly contradictory nature of the results, as stated by the authors in the first paragraph of Section 4.3. Let us ignore for now that the authors seem to ignore that prior work used the ensemble version of TabPFN, unlike the authors in their main experiments. Nevertheless, assuming (the non-ensembled version of) TabPFN has a large negative transfer for some datasets but is still dominant on all other datasets, this would result in TabPFN outperforming all other methods in rank and normalized score as reported in McElfresh et al. (2023) but might have a worse non-normalized score as reported by the authors in this work.

I highly recommend reporting ranks, at least. Moreover, please provide the raw results per dataset so that the reader can inspect them.
Ideally, reporting normalized scores following, for example, McElfresh et al. (2023), would also be highly beneficial to resolve any confounding factors or confusion caused by the analysis of the results.

- [1] Caruana, R., Niculescu-Mizil, A., Crew, G., & Ksikes, A. (2004, July). Ensemble selection from libraries of models. In Proceedings of the twenty-first international conference on Machine learning (p. 18).
- [2] Gijsbers, P., Bueno, M. L., Coors, S., LeDell, E., Poirier, S., Thomas, J., ... & Vanschoren, J. (2024). Amlb: an automl benchmark. Journal of Machine Learning Research, 25(101), 1-65.
- [3] Grinsztajn, L., Oyallon, E., & Varoquaux, G. (2022). Why do tree-based models still outperform deep learning on typical tabular data?. Advances in neural information processing systems, 35, 507-520.

## Clarification Required for Validation Data & Fine-tuning Data

The authors report using the 80-10-10 training-validation-test splits produced by McElfresh et al. (2023) as part of TabZilla but do not report how these were used.

I would assume that the 10% validation data were used for fine-tuning. But did the retrieval also use only the validation data? Moreover, when predicting the test instances, did the authors take all other data or only training data for retrieval? These questions deserve an answer in the paper as they are highly influential for joint retrieval and fine-tuning, which seems to dominate either approach.

On that note, I wondered if the authors used the pipeline by McElfresh et al. (2023) or custom code. If the former, then TabPFN [would not have used the validation data](https://github.com/naszilla/tabzilla/blob/main/TabZilla/models/tabpfn.py#L15-L18) at all, and all reported results use less data than the tuned alternatives; resulting in an unfair comparison. As far as I can tell, this problem already exists in the work by McElfresh et al. (2023).

Furthermore, related to comparing TabPFN:
* The authors use their own results of TabPFN, but for all other baselines, the results provided by McElfresh et al. (2023). How do the results of TabPFN obtained by McElfresh et al. (2023) compare here? Does the author's run of TabPFN reproduce the work by McElfresh et al. (2023) or perform worse? If the latter, why? This would seem to be quite important for the statements made by the authors in Section 4.3.
* The authors always compare their main results to the non-ensembled version of TabPFN (all results but Figure 7, right). Given that prior work and TabPFN, by default, use ensembling, what is the reasoning for this? This seems worth more details in the paper or simply using the ensembled version for the main results.

## Questions to Answer to Improve Clarity
* I am a bit unclear about Figure 4's setting and visualization of efficient local context computation for fine-tuning. As far as I understand, the figure is an example for B=5 but only for one batch, right? Adding this to the caption might help me understand the method better.
* Were the splits stratified when splitting into context and queries, Line 201, right column?
* An additional argument for 243ff. for kNN still being used in tabular machine learning could be that kNN is still used by automated machine learning systems such as AutoGluon.

## Misc
The authors make a very interesting point in Line 210, right column: "In practice, we observe that this method does not lead to any significant degradation in performance while allowing much faster training." I was wondering if the authors considered the following points for further investigation and if so, it would be great to include them in the paper.

Have the authors investigated decision boundaries related to the modified kNN search? It seems plausible that although "the neighbor of your neighbor might not be your neighbor," if your neighbor is close to a decision boundary, you are also close to the decision boundary.
The context might only be important/meaningful if we have a discrimination boundary within it (otherwise, there is nothing to learn for classification, although there might be something to learn for regression).

Conversely, have the authors considered optimization tricks like pruning batches without meaningful splits (context-queries combinations) during fine-tuning?

**Reason For Not Giving Higher Score:**

There are minor flaws in the experimental design and analysis of the results that might affect the conclusions.
Thus, without resolving them, I cannot recommend this work as a talk, given that its conclusions might not be entirely correct.

**Reason For Not Giving Lower Score:**

The paper is a perfect fit for the workshop, has a good novel idea, and has only some minor flaws.

---

### Official Review · Reviewer_8PHL · 2024-06-07
**Great paper, clearly written and with interesting findings and implications**

**Rating:** 3
**Fit:** 3
**Confidence:** 3

**Workshop Review:**

Context: I'm very familiar with the literature on tabular data and TabPFN, having read most of the papers referenced.

# Summary
The paper indicates that TabPFN struggles with high data complexity, a common trait in larger datasets. As a solution, the paper suggests using an "instance-wise" version of TabPFN, which is locally fitted on a subset of the training data nearest to the test point. Additionally, TabPFN is fine-tuned to enhance performance on local data subsets. The paper demonstrates that fitting TabPFN on local data results in a 2-5% performance improvement over TabPFN, with fine-tuning contributing an additional 2%. The final model slightly outperforms XGBoost.

# Strengths
- Very well-written and polished paper. It was an absolute joy to read it.
- Interesting findings that TabPFN cannot fit very complex data. A lot of work is currently being built on TabPFN, which is relevant for the community.
- Simple and effective method. I like the idea of using a local context, as proposed for TabPFN-KNN, because it's straightforward to implement—making it practical. However, I find the additional fine-tuning a bit impractical because it requires additional fine-tuning and eliminates the "magic” of TabPFN—which is that it's a training-free method.
- Well-designed experiments, with substantial ablations to decompose the effect of the individual model compoenents.
- The authors are honest about the work's limitations throughout the paper.

# Suggestions for improvements
- It's unclear how you encoded categorical data to work with TabPFN
- The fine-tuning required for the final 2% is quite cumbersome to obtain. There may be better ways to overcome the fine-tuning, especially to make it more practical (i.e., minimal coding required to still make TabPFN a one-click predictor).
- Expand the captions of the figures to explain the data used. For instance, the caption of Figure 8 should state that the numbers are from larger datasets.
- Line 215 left, the reference should be to Figure 4, not Section 2.4
- The paragraph between lines 370-376 doesn't add much and is not convincing.
- Consider adding ElasticNet to the table; it's usually very competitive on tabular data.

**Reason For Not Giving Higher Score:**

I am already positive about the paper.

**Reason For Not Giving Lower Score:**

N/A.

I think the paper is well-written and interesting for the community. It would be a great addition to the workshop.

---

### Meta-Review · Area_Chair_AS3K · 2024-06-17

**Recommendation:** 2

**Metareview:**

- good paper on relevant topic; a reviewer suggests some improvements to the eval

---

### Decision · Program_Chairs · 2024-06-17

Accept (Poster)